# Physical activity, sedentary behavior and pancreatitis risk: Mendelian randomization study

**Ruiqi Ling**[1], **Juan Liang**[2], **Shaojian Mo**[1], **Jiabao Qi**[1], **Xifeng Fu**[1], **Yanzhang Tian**[1]*

**1** General Surgery Department, Shanxi Bethune Hospital/General Surgery Department, Third Hospital of Shanxi Medical University, Taiyuan, China, **2** The First Clinical Medical School, Shanxi Medical University, Taiyuan, China

* tyz2030@163.com

## Abstract

**Data Availability Statement:** All the data comes from online databases, The patients/participants provided their written informed consent to participate in this study. All data can be found in

### Background

Although observational studies have shown that physical activity is a protective factor for acute pancreatitis, the causal associations between PA/ sedentary behavior and acute pancreatitis (AP) and chronic pancreatitis (CP) remain unclear.

### Methods

We used Mendelian randomization as a strategy to assess the causalities between exposures and outcomes by simulating randomized experiments with genetic variation. The collected genetic variants data of physical activity were from UK Biobank, the data on sedentary behavior were also from UK Biobank, and both of them could be found in the GWAS catalog, and the data on AP and CP were from FinnGen. There were three physical activity related activity patterns (moderate to vigorous physical activity [MVPA], accelerometer-based physical activity with average acceleration, [AccAve] and accelerometer-based physical activity with accelerations >425 milli-gravities, [Acc425]) and three sedentary behavior-related lifestyle patterns (Leisure screen time [LST], Sedentary commuting, Sedentary behavior at work). We used inverse variance weighted (IVW), weighted median and MR-Egger for the analysis of Mendelian randomization, followed by sensitivity tests with the Cochran Q test, MR-Egger intercepts analysis and MR-PRESSO.

### Results

A causal relationship was found between LST and acute pancreatitis based on IVW analysis (odds ratios [OR] = 1.38, corresponding 95% confidence intervals [CI] = 1.16–1.64, p = 0.0002) and there were no causal relationships between physical activity/sedentary behavior and chronic pancreatitis. Sensitivity analysis showed no pleiotropy and heterogeneity of the results.

the Data sources section of the paper or Supporting Information as well as links to the data.

**Funding:** Our research received funding from the Shanxi Provincial Department of Human Resources and Social Security (grant no. 20210002) and the Shanxi Provincial Department of Science and Technology (grant no. 202104041101024). There was no additional external funding received for this study. Furthermore, we would like to clarify that the funders had no involvement in the study design, data collection, and analysis.

**Competing interests:** The authors declare no conflict of interest.

## Conclusions

Results show that reducing LST contributes to the prevention of acute pancreatitis, thereby reducing the health burden associated with it.

## Introduction

Physical activity is a vital part of people's lives and is closely related to healthy [1]. Along with lifestyle and working changes, physical activity is gradually decreasing and other insalubrious lifestyles such as sedentary behavior are increasing. The rapid global movement of coronavirus disease 2019 (COVID-19) has further aggravated this problem [2]. New evidence has been shown physical activity helps prevent tumors [3], diabetes [4], cardiovascular disease [5], mental disorders [6] and other diseases [7]. Sedentary behavior is characterized by low energy expenditure activities (metabolic equivalents [METs] $< 2.0$), mostly in a sitting or supine position [8]. It has been confirmed to be a risk factor for obesity [9], autoimmune disease [10] and cancer [11], which has become a health burden and even influence mortality [12].

AP is the most common Gastrointestinal emergency in advanced countries [13]. The course of acute pancreatitis is difficult to predict despite various assessment criteria [14]. Although with the diversification of treatments, approximately 20% of patients develop moderate to severe cases and even die [13]. More than 3% of cases of acute pancreatitis progress to chronic pancreatitis within 3–8 years [15]. CP is a fibroinflammatory syndrome of the exocrine pancreas. With increasing pain, leading to diabetes and a higher risk of cancer, chronic pancreatitis greatly deteriorates life quality [15, 16]. In order to reduce the health burden of pancreatitis, early prevention is essential. Smoking and alcohol consumption are recognized as important risk factors for both acute and chronic pancreatitis. Reducing tobacco and alcohol intake is one of the key preventive measures for lowering the risk of pancreatitis [17]. A recent observational study suggests that increasing the frequency of physical activity reduces the risk of acute pancreatitis and is not related to whether physical activity is occupationally related [18]. Physical activity may modulate the endocrine effects of the pancreas, thereby reducing the risk of pancreatitis [19]. However, there is still a void regarding the causal relationship between physical activity/sedentary behavior and pancreatitis.

Mendelian randomization (MR) is an important strategy to demonstrate causality between exposure and outcome [20]. Instrumental variables used by MR to evaluate exposure-outcome relationships is genetic variants, Statistical methods for minimizing the impact of confounders [21]. Genetic variations also influence the development of pancreatitis. At conception, genetic variants are randomly assigned and are not subject to disease progression, so reverse causality can be largely avoided [22]. The evidence level of randomized controlled trials (RCT) is highest in evidence-based medicine, but for ethical and economic reasons, RCT is generally feasible [23]. In the absence of RCT, MR has a higher level of evidence than observational studies. The purpose of this study was to reveal whether physical activity/sedentary behavior is causally related to pancreatitis based on the results of two samples of MR scanning.

## Methods

### Data sources

**Instrumental variable selection data.** The AP genome-wide association study (GWAS) data which has more than 377,000 participants was from UK Biobank [24]. Three of

phenotypes were selected, including moderate to MVPA, AccAve and Acc425. A questionnaire that refers to the international physical activity questionnaire (August 2002 version) was used to assess the physical activity, those moderate and high activity groups were classified into MVPA (N = 377,324) [24, 25]. Accelerometer-based physical activity required participants to wear an Axivity AX3 accelerometer for at least 72h in a week [26]. There are two categories of data measured from accelerometer which were classified as AccAve (N = 91,084) and Acc425 (N = 90,667) [24]. GWAS data for sedentary behavior were referenced from a recent high-quality meta-analysis that included GWAS data from 51 studies in which we selected data on sedentary behavior of European ancestry [27]. The phenotypes were classified as LST (N = 526,725), Sedentary commuting (N = 159,606), Sedentary behavior at work (N = 372,609). Detailed definitions of the three phenotypes can be found from the supplementary data in Zhe Wang study [27]. The data can be found on GWAS catalog (https://www.ebi.ac.uk/gwas/).

**Data sources for acute and chronic pancreatitis.**    The acute/chronic pancreatitis datasets we used are from FinnGen (https://r8.finngen.fi/), all data were updated in December 2022. The reason we used FinnGen datasets is to rule out the possibility of overlapping samples which may cause Sample overlap bias. There are 5,509 cases and 301,383 controls in the statistics of acute pancreatitis data. We also collected 3,002 cases and 301,383 controls for chronic pancreatitis, all participants own ancestry of European.

## Study design

Exposure related single-nucleotide polymorphisms (SNPs) were used as IVs in MR study (Fig 1). To perform MR analysis, three assumptions must be met by IVs (Fig 2): 1. There is a strong association between exposure and SNPs; 2. There should be no association between SNPs and potential confounders 3. SNPs have something to do with outcome only through exposure [20]. SNPs that were significantly ($P < 5 \times 10^{-8}$) associated with physical activity/sedentary behavior were selected, after we clumped these SNPs in linkage disequilibrium (LD, Clumping window: 10000kb, Clumping $r^2$ cutoff: 0.001) and used Pheno Scanner V2 (http://www.phenoscanner.medschl.cam.ac.uk/) to eliminate confounding SNPs (smoking and alcohol use related). Smoking and alcohol use are confounding factors, and if the selected instrumental variables are associated with them, the study would not meet the second assumption of Mendelian randomization (MR). Therefore, it is important to exclude SNPs that are associated with smoking and alcohol use to avoid confounding factors affecting the causal relationship. For phenotypes with fewer than 3 SNPs, we rescreened with a relaxed P value of less than $5 \times 10^{-7}$, all other screening conditions unchanged [28]. The final SNPs of MVPA, AccAve, Acc425, LST, Sedentary commuting and Sedentary behavior at work are 16, 7, 4, 92, 3, 6 and all SNPs F-statistics >10 (The S1-S6 Tables in S1 Appendix provides details about SNPs).

## Mendelian randomization and sensitivity analysis

The causal effects of physical activity/sedentary behaviors on AP and CP were analyzed by IVW method, weighted median with MR-Egger was used as a supplement. If there were no heterogeneity and pleiotropy in IVs, the fixed-effects model analysis using IVW was the most accurate, but when heterogeneity was present, the random-effects model of IVW or methods of weighted median is preferred. In addition, It is suggested to use MR-Egger analysis instead of IVW and weighted median in the case of pleiotropy of SNPs [29]. Cochran Q test was performed to test for heterogeneity and the funnel plot is used as a supplement. If instrumental variables exert a direct effect on the outcome independent of the exposure factor, it violates the fundamental principle of Mendelian randomization and suggests the presence of horizontal

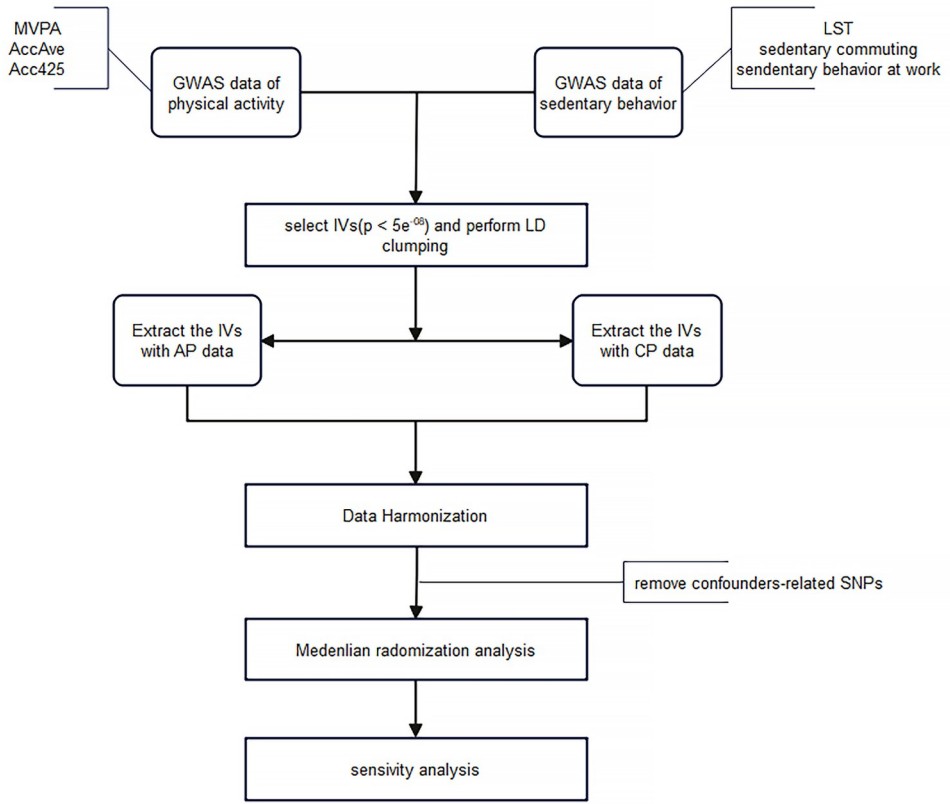

**Fig 1. The process of instrumental variable selection and study design.** IVs: Instrumental variables; MVPA: moderate to vigorous physical activity; AccAve: accelerometer-based physical activity (average acceleration); Acc425: accelerometer-based physical activity (fraction of time with accelerations >425 milli-gravities); LST: leisure screen time; AP: acute pancreatitis; CP: chronic pancreatitis.

pleiotropy. To avoid potential biases arising from the presence of horizontal pleiotropy in causal relationships, we employ MR-Egger intercept analysis to assess the horizontal pleiotropy in MR results. MR pleiotropy residual sum and outlier (MR-PRESSO) was conducted for

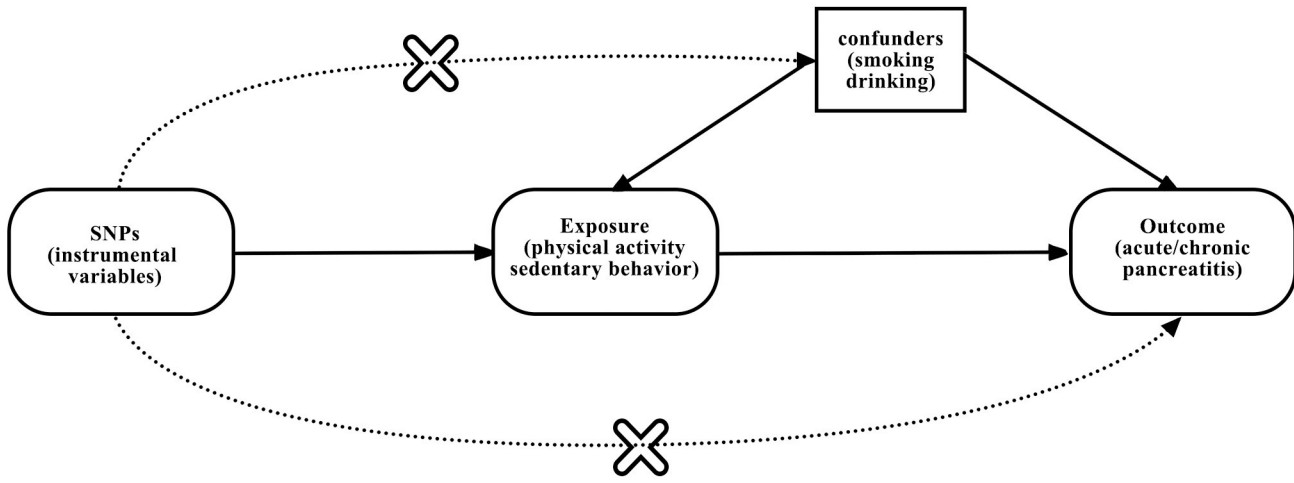

**Fig 2. Instrumental variables fit the assumptions of MR.** SNPs: single-nucleotide polymorphisms.

assessing vertical pleiotropy and detection of biased SNPs (p < 0.05). MR-PRESSO was not feasible when there were less than or equal to three SNPs. For the presence of biased SNPs, MR and sensitivity analysis were repeated after excluding bias. At last, leave one out analysis was used to assess the robustness of the analysis results [30].

## Statistical analysis

All statistical analyses were performed in R 4.2.2 using the R package TwoSampleMR (version 0.5.6). Because multiple independent analyses were performed, the P value (p < 0.0083: 0.05/6) after Bonferroni correction was used as the criterion for significance. The results of MR Analysis were shown in odds ratios (OR) with corresponding 95% confidence intervals (CI). Statistical power was calculated using mRnd(https://shiny.cnsgenomics.com/mRnd/)

## Ethics statement

All the data comes from online databases, The patients/participants provided their written informed consent to participate in this study.

# Results

## Physical activity/sedentary behavior and AP

Inverse variance weighted (IVW) Analysis showed that only LST had a causal relationship with acute pancreatitis (OR = 1.38, 95CI% = 1.16–1.64, p = 0.0002) and the statistical power was 0.99. Two other sedentary behavior-related phenotypes have been shown to reduce the risk of AP(Fig 3). Other outcomes indicated that all three indicators of physical activity reduced the risk of acute pancreatitis but without causal collections. (MVPA: OR = 0.45, 95CI% = 0.19–1.06, p = 0.0682; AccAve: OR = 0.96, 95CI% = 0.89–1.03, p = 0.2737; Acc425: 0.40, 95CI% = 0.14–1.18, p = 0.0980). The results of weighted median and MR-Egger were listed in S7 Table (S1 Appendix).

## Physical activity/sedentary behavior and CP

In total, neither the indicators of physical activity nor sedentary behavior were causally linked to CP. MVPA, AccAve and Acc425 play active roles for prevention of CP in IVW analysis (Fig 4). LST and Sedentary commuting increased the risk of CP but Sedentary behavior at work reduced the risk. Weighted median and MR-Egge results also can be found in S7 Table (S1 Appendix).

## Sensitivity analyses

To further evaluate the results' precision, sensitivity analyses, including MR-Egger intercept analysis, Cochran Q test, and MR-PRESSO global test, were conducted. MR-Egger intercept and MR-PRESSO analysis indicated that the pleiotropy of result of LST to AP/CP doesn't exist (Fig 5) and we can draw robust conclusions after examining (S1 Fig). In other results, there was no horizontal pleiotropy since all MR-Egger intercepts were p > 0.05. The results of Cochran Q test indicated the heterogeneity of results when studying the relationship between AccAve and CP (Q = 15.15, p = 0.02) and MR-PRESSO shwon rs34517439 biased the results. After we eliminated rs34517439, we re-performed MR Analysis and sensitivity analysis of AccAve and CP. This time, Cochran Q test (Q = 6.66, p = 0.25) and MR-PRESSO global test (p = 0.28) shown that there is no heterogeneity in the results. There was no either heterogeneity nor pleiotropy in the other results (Table 1).

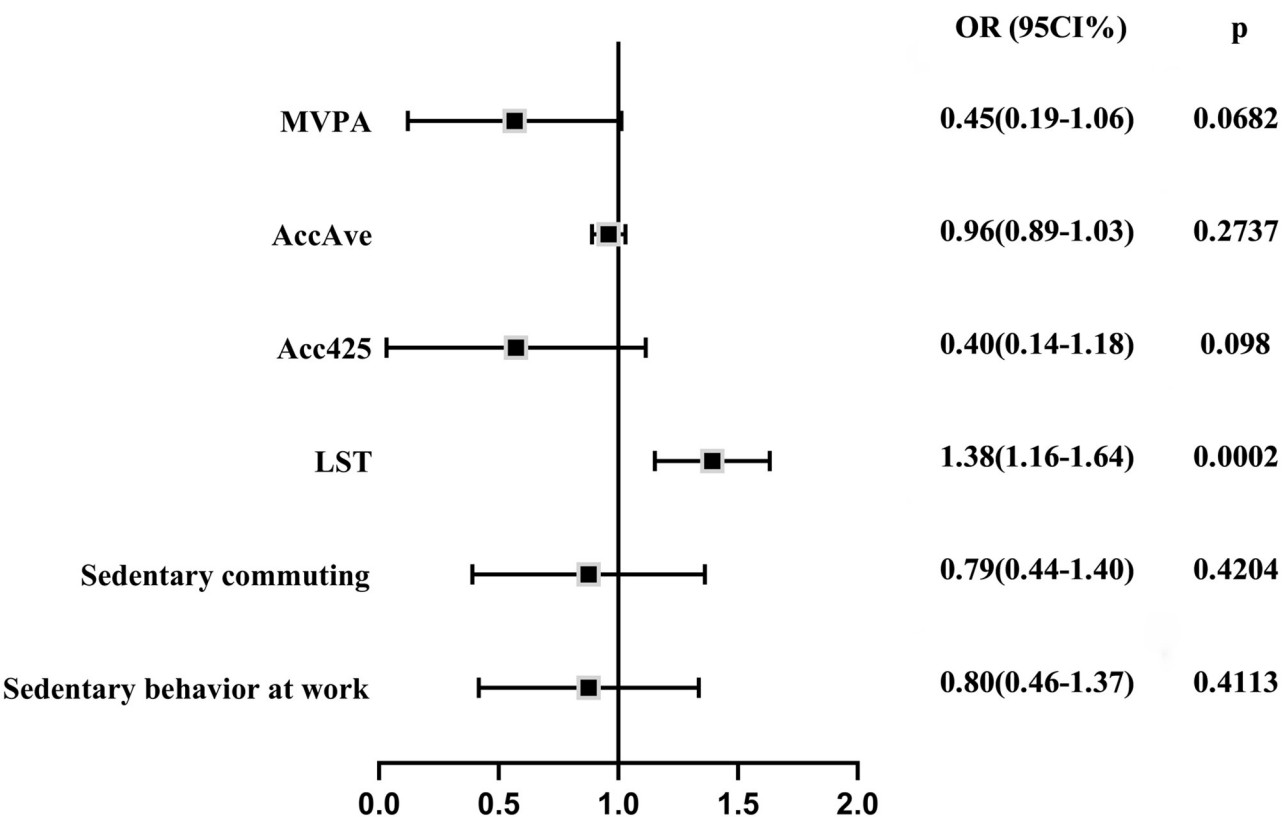

**Fig 3. Causal effects of physical activity/sedentary behavior on acute pancreatitis by IVW.** MVPA: moderate to vigorous physical activity; AccAve: accelerometer-based physical activity (average acceleration); Acc425: accelerometer-based physical activity (fraction of time with accelerations >425 milli-gravities); LST: leisure screen time; IVW: inverse variance weighted.

## Discussion

To date, no studies have investigated the causal relationship between physical activity/sedentary behavior and AP/CP.

Physical activity and sedentary behavior are the two most common lifestyle habits of people. As people pay more attention to health, research on physical activity and sedentary behavior has exploded in recent years. Although latest studies have identified that physical activity/sedentary behavior is associated with cancer [31], various chronic diseases, and even some studies have shown that sedentary behavior can increase the severity of COVID-19 and the probability of hospitalization. However, studies on their association with gastrointestinal diseases are still scarce.

Pancreatitis is one of the most common gastrointestinal diseases. There was a reduction in the risk of AP among physical activity participants in a large observational study in China. Observational studies, however, are susceptible to confounding factors and reverse causality, and Asian findings may not generalize to European populations. There are even fewer studies on physical activity/sedentary behavior and CP, and these research directions have focused on influence of prognosis. It is still difficult for patients with acute pancreatitis and chronic pancreatitis to have a good quality of life during their illness, despite major improvements in treatment. Therefore, the prevention of pancreatitis is important and understanding the protective and risk factors of pancreatitis is urgent.

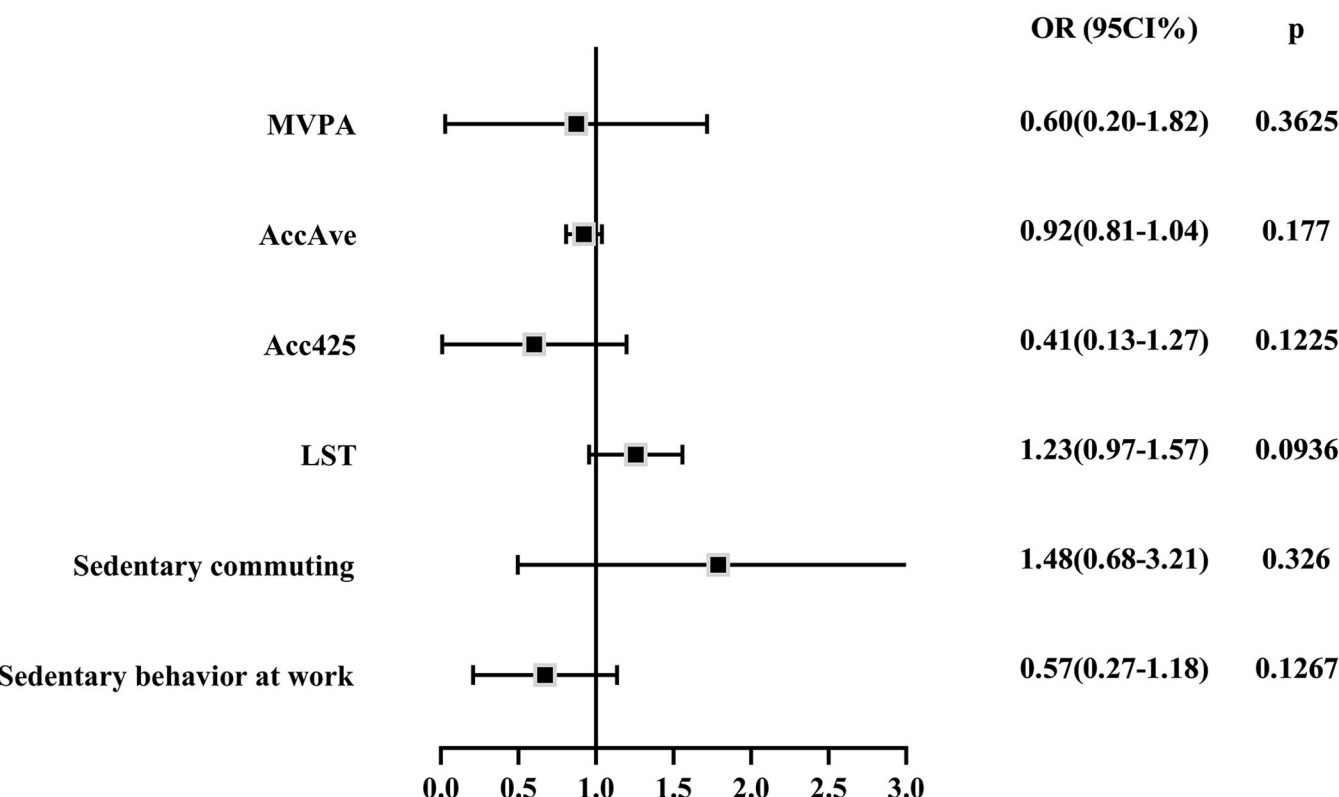

**Fig 4. Causal effects of physical activity /sedentary behavior on acute pancreatitis by IVW.** MVPA: moderate to vigorous physical activity; AccAve: accelerometer-based physical activity (average acceleration); Acc425: accelerometer-based physical activity (fraction of time with accelerations >425 milli-gravities); LST: leisure screen time; IVW: inverse variance weighted.

It was our study that first demonstrated causally that LST increases AP risk. It may caused by following reasons. Firstly, LST people often have abnormal eating habits, they usually don't have a habit of breakfast and the intake of vitamin and dietary fiber in the diet is low, fat composition is relatively high [32].These irregular eating habits can induce the formation of gallstones [33] and gallstones is the most common trigger of AP in advanced countries [34]. Secondly, A recent Mendelian randomization study has demonstrated that LST increases the risk of type 2 diabetes causally [35]. Dipeptidylpeptidase-4 (DPP4) inhibitors and Glucagon-like peptide-1 receptor agonist (GLP-1RAs) are important treatments for T2D.These medicines may increase the risk of pancreatitis [36, 37]. Last but not least, sedentary behavior will increase risk of autoimmune disease [10] which also a risk factor for AP in European populations [13]. In our study, it was indeed not shown that physical activity would affect AP/CP. This contradicts the findings of previous observational study [18]. We speculate that this may caused by people who exercise regularly will have less bad lifestyle habits such as smoking and drinking. Smoking and drinking are risk factors for pancreatitis which cause the bias [16]. Qilin Qian's study confirmed that physical activity reduces the risk of gallstone disease [38], which may cause bias of observational study.

We used SNPs instead of exposure to investigate the causal relationship between exposures and outcomes. Compared with observational studies, this strategy can effectively avoid interference of confounders and reverse causality. In the selection of GWAS data, we selected data with European ancestry, which can largely avoid the bias caused by different races. Different databases were used for the selection of GWAS data for exposures and outcomes to avoid bias

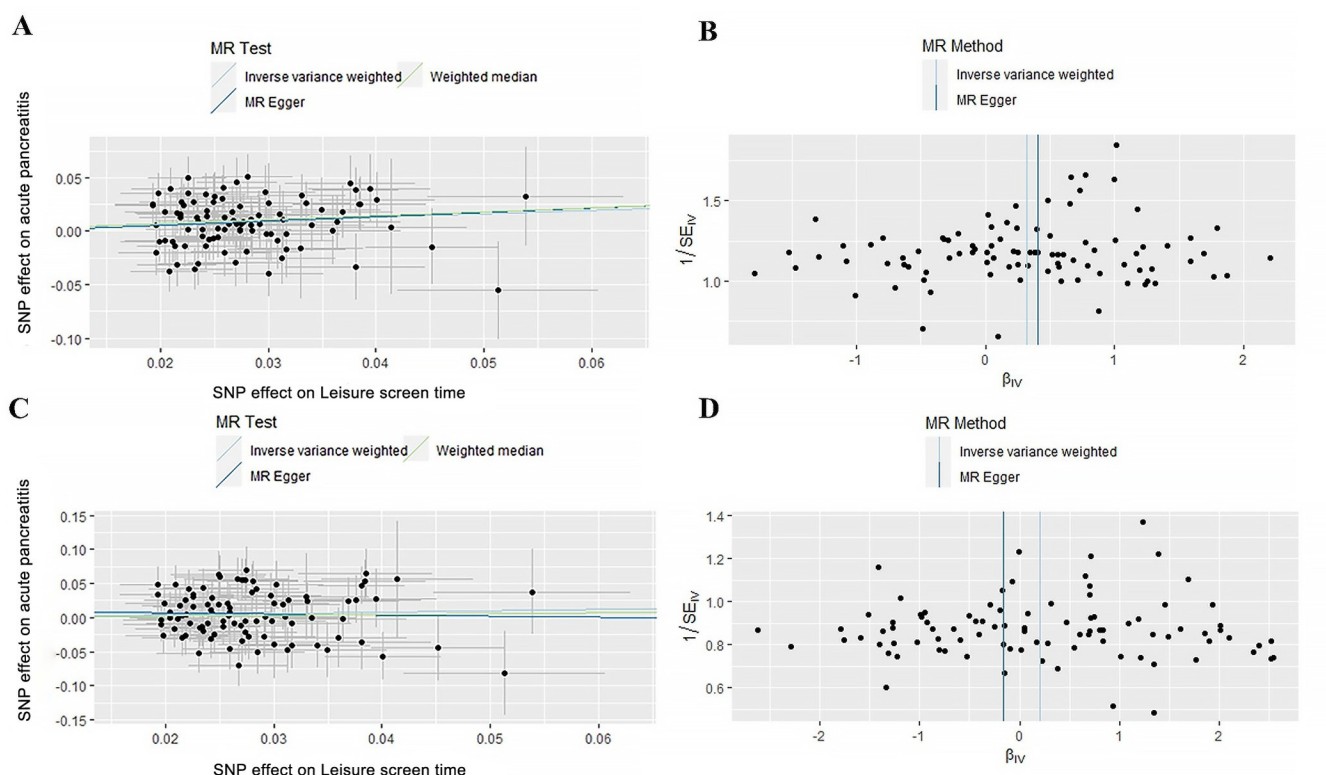

**Fig 5. Sensitivity analyses of MR results about LST to acute/chronic pancreatitis.** a, b: Scatter plots and funnel plots show genetically predicted of LST on acute pancreatitis; c, d: genetically predicted LST on chronic pancreatitis. LST: Leisure screen time.

due to overlapping samples. Our results have clinical implications, and reducing LST can effectively reduce the risk of AP, thereby reducing the health burden of European countries.

However, our study still has some limitations. In the first place, because all GWAS data were derived from participants of European ancestry, our results only apply to European

**Table 1. Sensitivity analysis of the causal association between physical activity/sedentary behavior and the risk of AP/CP.**

| Exposure | Outcome | MR-Egger | | Cochran Q test | | MR-PRESSO |
|---|---|---|---|---|---|---|
| | | Intercept | p | Q value | P | p |
| MVPA | AP | 0.04 | 0.38 | 20.44 | 0.16 | 0.17 |
| | CP | -0.03 | 0.54 | 19.39 | 0.20 | 0.20 |
| AccAve | AP | -0.06 | 0.21 | 4.06 | 0.67 | 0.68 |
| | CP | -0.06 | 0.44 | 6.66 | 0.25 | 0.28 |
| Acc425 | AP | -0.59 | 0.18 | 4.82 | 0.19 | 0.27 |
| | CP | -0.32 | 0.50 | 1.00 | 0.80 | 0.77 |
| LST | AP | -0.002 | 0.85 | 86.47 | 0.61 | 0.62 |
| | CP | 0.01 | 0.53 | 99.67 | 0.25 | 0.25 |
| Sedentary commuting | AP | 0.02 | 0.86 | 0.29 | 0.86 | - |
| | CP | -0.02 | 0.91 | 0.47 | 0.79 | - |
| Sedentary behavior at work | AP | -0.03 | 0.46 | 3.62 | 0.61 | 0.63 |
| | CP | -0.04 | 0.43 | 3.50 | 0.62 | 0.67 |

MVPA: moderate to vigorous physical activity; AccAve: accelerometer-based physical activity (average acceleration); Acc425: accelerometer-based physical activity (fraction of time with accelerations >425 milli-gravities); LST: leisure screen time; AP: acute pancreatitis; CP: chronic pancreatitis.

populations and cannot be generalized to all humans. Moreover, detailed disease information was not available for participants in the course of the GWAS data collection, stratified analyses could not be performed and it is not possible to infer whether physical activity/sedentary behavior has a causal link to disease severity in pancreatitis either. In addition, except for the MR estimates of LST and AP, the statistical power of the other MR Analysis results was low which may lead to false negative results.

All in all, there are no causal relationship between physical activity and pancreatitis. It's LST rather than sedentary behavior increase the risk of AP causally but it has no causal connection with CP. This suggests that reducing the amount of time spent on electronics may reduce the risk of acute pancreatitis.

## Supporting information

**S1 Appendix. S1-S7 Tables are included in file.**
(XLS)

**S1 Fig.**
(DOCX)

## Acknowledgments

We thank all participants and investigators of the UK Biobank and the FinnGen study.

## Author Contributions

**Data curation:** Ruiqi Ling, Juan Liang, Shaojian Mo.

**Formal analysis:** Ruiqi Ling.

**Methodology:** Jiabao Qi.

**Project administration:** Yanzhang Tian.

**Supervision:** Xifeng Fu.

**Visualization:** Ruiqi Ling.

**Writing – original draft:** Ruiqi Ling.

**Writing – review & editing:** Ruiqi Ling.

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
