## [Decision Letter · Decision Letter 0]

19 May 2023

PONE-D-23-04358physical activity, sedentary behavior and pancreatitis risk: Mendelian randomization studyPLOS ONE

Dear Dr.Tian,

Thank you for submitting your manuscript to PLOS ONE. After careful consideration, we feel that it has merit but does not fully meet PLOS ONE’s publication criteria as it currently stands. Therefore, we invite you to submit a revised version of the manuscript that addresses the points raised during the review process.

ACADEMIC EDITOR:Dear Author, Please attend to all the reviewers' comments and make necessary corrections. The decision of this manuscript is justified based on PLOS ONE’s publication criteria and not on its novelty or perceived impact.

We look forward to receiving your revised manuscript.

Kind regards,

Zulkarnain Jaafar

Academic Editor

PLOS ONE

Journal Requirements:

Reviewers' comments:

Reviewer's Responses to Questions

**Comments to the Author**

1. Is the manuscript technically sound, and do the data support the conclusions?

Reviewer #1: Yes

Reviewer #2: Yes

2. Has the statistical analysis been performed appropriately and rigorously? 

Reviewer #1: No

Reviewer #2: Yes

3. Have the authors made all data underlying the findings in their manuscript fully available?

Reviewer #1: Yes

Reviewer #2: Yes

4. Is the manuscript presented in an intelligible fashion and written in standard English?

Reviewer #1: Yes

Reviewer #2: Yes

5. Review Comments to the Author

Reviewer #1: My dear authors ;

Many thanks for great work and discuss very important risk factors like sendenttary life or physical activity

the strenght points :

1- The strength of the present study is the MR design, which minimized bias from

confounding and reverse causality and use many sensitivity analyses to avoid bias

whats expectations from authors

1- how to calculate sample size in the study

2- whats number of instruments is used for MVPA as small number may affect on the power of study

3- please write about Pleitropic effects of smoking and alcohol consumption and data sources as paragraph

4- whats about the effect of horizanatal pleiotropic on bias

5- you use three indicators for sedentary behaviour and three indicators for physical activity

6- at conclusion please write about the linking between physcial activity and AP and CP is determined or not

7- Are individuals of the UK Biobank were included in both the exposure and outcome datasets as this potential limitation which might introduce some bias in the causal estimates in the direction of the estimates of observational studies

8- you should give titles regarding the data sources as following :

*instrumentalvariable selection data

*data sources for acute and chronic pancreatitis

*data for smoking and drinking alcohol

9- why no figure shown illustating the study design and Study design to explain easily for the readers

10- why use abbreviation for physical activity i think not good this abbreviations as not specific

Reviewer #2: This study is interesting. However, there are still some parts to be improved such as the figure 4 presentation (blurred), punctuation, and abstract writing.

1. Please enhance the result section in abstract with other important findings.

2. line 68-69 In which working system in the body the PA could reduce the risk for AP? Please explore this explanation in this paragraph concisely.

3. line 73 Do the genetic variants are chosen based on the genetic that involve in the pancreatitis development?

4. line 87 Mention the name of the questionnaire used in this study

5. line 221-222 Why author did not use the dietary intake data of each subject to be one of their co-variables?

6. line 251 Delete the dot (.)

7. line 252-254 Please add one sentence to inform the public recommendation based on the study findings.

6. PLOS authors have the option to publish the peer review history of their article (what does this mean?). If published, this will include your full peer review and any attached files.

Reviewer #1: No

Reviewer #2: **Yes: **Arif Sabta Aji

---

## [Author Response · Author response to Decision Letter 0]

25 May 2023

Dear academic editor and reviewers,

Thank you very much for your comments and professional advice These opinions help to improve the academic rigor of our article Based on your suggestion and request we have made corrected modifications to the revised manuscript. We hope that our work can be improved again Furthermore, we would like to show the details as follows:

Academic editor:

5.Please review your reference list to ensure that it is complete and correct. If you have cited papers that have been retracted, please include the rationale for doing so in the manuscript text, or remove these references and replace them with relevant current references. Any changes to the reference list should be mentioned in the rebuttal letter that accompanies your revised manuscript. If you need to cite a retracted article, indicate the article’s retracted status in the References list and also include a citation and full reference for the retraction notice.

The author's answer:

1.We have followed the magazine's requirements for the author information and title format in the manuscript and deleted other unnecessary sections to make sure that our manuscript meets PLOS ONE's style requirements.

2.2.We kindly request to modify our financial disclosure statement. Our research received funding from the Shanxi Provincial Department of Human Resources and Social Security (grant no. 20210002) and the Shanxi Provincial Department of Science and Technology (grant no. 202104041101024). The funding agency is represented by the abbreviation YT. Furthermore, we would like to clarify that the funders had no involvement in the study design, data collection, and analysis.

3.We have moved the ethics statement to the Methods section and deleted it from other section

4.We have added Supporting Information files at the end of oour manuscript and updated any in-text citations to match accordingly

5.We added a reference, which is the 19th reference in the full text, to add the rationale for our study. The reference list is complete and correct. We have not cited papers that have been retracted

Reviewer #1: 

1- how to calculate sample size in the study

2- whats number of instruments is used for MVPA as small number may affect on the power of study

3- please write about Pleitropic effects of smoking and alcohol consumption and data sources as paragraph

4- whats about the effect of horizanatal pleiotropic on bias

5- you use three indicators for sedentary behaviour and three indicators for physical activity

6- at conclusion please write about the linking between physcial activity and AP and CP is determined or not

7- Are individuals of the UK Biobank were included in both the exposure and outcome datasets as this potential limitation which might introduce some bias in the causal estimates in the direction of the estimates of observational studies

8- you should give titles regarding the data sources as following :

*instrumentalvariable selection data

*data sources for acute and chronic pancreatitis

*data for smoking and drinking alcohol

9- why no figure shown illustating the study design and Study design to explain easily for the readers

10- why use abbreviation for physical activity i think not good this abbreviations as not specific

The author's answer:

1: We use data from UK Biobank and FinnGen as our sources. The sample size in the database remains constant, and in order to achieve greater statistical power, we have included all eligible data that meets the criteria. The calculation of sample size and statistical power is performed using the website https://shiny.cnsgenomics.com/mRnd/. (page5 section of Instrumental variable selection data) 

2: The number of instrumental variables (IVs) we have selected for MVPA is 16, which were carefully screened through strict selection steps. This quantity is not small and is similar to the number of MVPA experiments used as instruments in previous Mendelian randomization studies. However, as our outcome differs from those studies, there may be slight variations in the number of instrumental variables used. The statistical power we calculated of MVPA to acute/chronic pancreatitis were 0.32 and 0.12(page6 line 126-128)

3: Smoking and alcohol use are confounding factors, and if the selected instrumental variables are associated with them, the study would not meet the second assumption of Mendelian randomization (MR). Therefore, it is important to exclude SNPs that are associated with smoking and alcohol use to avoid confounding factors affecting the causal relationship. They only play confounding roles, so the data on smoking and alcohol use is unnecessary.(page 3 line 63-65)

4. we have added the effect of horizontal pleiotropic on bias(page7 line 142-147)

5: Regarding the selection of three phenotypes related to physical activity and sedentary behavior, we referred to all relevant GWAS and Mendelian randomization studies. We chose the three phenotypes that were most frequently used and representative in these studies as indicators for our analysis(page 5 line89-92)

6: physical activity has no causal relationship with acute/chronic pancreatitis, which we have added in the paper.(page8 line169-171)

7: the exposure date came from UK biobank and the outcome date was from FinnGen, They came from different countries so there was no bias due to sample overlap.(page2 line24-27)

8: we have reversed titles regarding the data sources, but the data didn't include data for smoking and drinking alcohol. Smoking and drinking alcohol only play confounding roles. And we ticked out the SNPs associated with them to make sure our MR won't be interfered by confounders.(page4-5 section of data sources)

9: We have added a figure(Fig 1) illustrating the study design and Study design to explain easily to the readers(page6 line131)

10: this abbreviation was not specific, so we have canceled the abbreviate physical to PA.

Reviewer #2: 

This study is interesting. However, there are still some parts to be improved such as the figure 4 presentation (blurred), punctuation, and abstract writing.

1. Please enhance the result section in abstract with other important findings.

2. line 68-69 In which working system in the body the PA could reduce the risk for AP? Please explore this explanation in this paragraph concisely.

3. line 73 Do the genetic variants are chosen based on the genetic that involve in the pancreatitis development?

4. line 87 Mention the name of the questionnaire used in this study

5. line 221-222 Why author did not use the dietary intake data of each subject to be one of their co-variables?

6. line 251 Delete the dot (.)

7. line 252-254 Please add one sentence to inform the public recommendation based on the study findings.

The author's answer:

We have adjusted our picture4 to make it won’t blur(We added an additional figure, so Figure 4 is now referred to as Figure 5.) 

1: We have enhanced the result section in the abstract by adding the relationship between physical activity/sedentary behavior and chronic pancreatitis.(page2 line 35-39)

2: Physical activity may modulate the endocrine effects of the pancreas, thereby reducing the risk of pancreatitis.(page 4 line 68-69)

3: Genetic variations also influence the development of pancreatitis.(page 4 line 75-76)

4: The questionnaire referred to is the international physical activity questionnaire (August 2002 version)(page5 89-92)

5: There are 2 reasons to explain why we didn't use the dietary intake data of each subject to be one of their co-variables (page12 line 240-242)

1. the data we collected do not have the dietary intake data. The database builders did not consider the collection of this part of the data.

2. The SNPs selected as the instrument variable do not have connections with dietary intake.

6: We have deleted the dot (.)(page13 line 271-273)

7: We have added one sentence to inform the public recommendation based on the study findings.(page13 line 275-276)

---

## [Decision Letter · Decision Letter 1]

13 Jun 2023

physical activity, sedentary behavior and pancreatitis risk: Mendelian randomization study

PONE-D-23-04358R1

Dear Dr. Tian,

We’re pleased to inform you that your manuscript has been judged scientifically suitable for publication and will be formally accepted for publication once it meets all outstanding technical requirements.

Kind regards,

Zulkarnain Jaafar

Academic Editor

PLOS ONE

Additional Editor Comments (optional):

Reviewers' comments:

Reviewer's Responses to Questions

**Comments to the Author**

1. If the authors have adequately addressed your comments raised in a previous round of review and you feel that this manuscript is now acceptable for publication, you may indicate that here to bypass the “Comments to the Author” section, enter your conflict of interest statement in the “Confidential to Editor” section, and submit your "Accept" recommendation.

Reviewer #1: All comments have been addressed

Reviewer #2: (No Response)

2. Is the manuscript technically sound, and do the data support the conclusions?

Reviewer #1: Yes

Reviewer #2: Yes

3. Has the statistical analysis been performed appropriately and rigorously? 

Reviewer #1: I Don't Know

Reviewer #2: Yes

4. Have the authors made all data underlying the findings in their manuscript fully available?

Reviewer #1: Yes

Reviewer #2: Yes

5. Is the manuscript presented in an intelligible fashion and written in standard English?

Reviewer #1: Yes

Reviewer #2: Yes

6. Review Comments to the Author

Reviewer #1: My dear authors all comments have been addressd

Many thanks for big efforts but at answers for reviewers the uthor remembered the magazine requirments so it is better to called journal not magazine

Reviewer #2: Authors have already revised all the comment. We suggest to continue the process and accept this manuscript.

7. PLOS authors have the option to publish the peer review history of their article (what does this mean?). If published, this will include your full peer review and any attached files.

Reviewer #1: **Yes: **Amr ahmed

Reviewer #2: **Yes: **Arif Sabta Aji

---

## [Editor Report · Acceptance letter]

11 Jul 2023

PONE-D-23-04358R1 

Physical activity, sedentary behavior and pancreatitis risk: Mendelian randomization study 

Dear Dr. Tian:

I'm pleased to inform you that your manuscript has been deemed suitable for publication in PLOS ONE. Congratulations! Your manuscript is now with our production department. 

Kind regards, 

on behalf of

Dr. Zulkarnain Jaafar 

Academic Editor

PLOS ONE